# Effects of Planting Date for Soybean Growth, Development, and Yield in the Southern USA

**Nick R. Bateman [1],\*, Angus L. Catchot [2],\*, Jeff Gore [3], Don R. Cook [3], Fred R. Musser [2]** 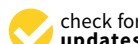 **and J. Trent Irby [4]**

[1] Department of Entomology and Plant Pathology, University of Arkansas Cooperative Extension Service, Rice Research and Extension Center, 2900 Highway 130 E, Stuttgart, AR 72160, USA
[2] Department of Biochemistry, Molecular Biology, Entomology, and Plant Pathology, Mississippi State University, 100 Old Highway 12, Starkville, MS 39762, USA; fmusser@entomology.msstate.edu
[3] Department of Biochemistry, Molecular Biology, Entomology and Plant Pathology, Mississippi State University, Delta Research and Extension Center, 82 Stoneville Rd P.O. Box 197, Stoneville, MS 38776, USA; jgore@drec.msstate.edu (J.G.); dcook@drec.msstate.edu (D.R.C.)
[4] Department of Plant and Soil Sciences, Mississippi State University, 117 Dorman Hall Starkville, MS 39762, USA; trent.irby@msstate.edu
\* Correspondence: nbateman@uaex.edu (N.R.B.); acatchot@entomology.msstate.edu (A.L.C.)

**Abstract:** As fluctuating commodity prices change the agriculture landscape on a yearly basis, soybean (*Glycine max* (L.) Merr.) has become the predominant crop in the southern USA, accounting for 65 percent of the total row crop production in the state. To accommodate increased soybean production, planting dates have expanded, spanning from late March through July. To determine the impact of this expanded planting window on soybean development and yield, field experiments were conducted at Starkville and Stoneville, MS, in 2013 and 2014. Treatments included seven planting dates ranging from 25 March to 15 July and two soybean cultivars (one Maturity Group IV and one Maturity Group V cultivar). These studies were conducted in irrigated high—-yielding environments. Experimental units were sampled weekly for insect pests and insecticides were applied when populations exceeded the levels at which applications were recommended. Planting date had a significant impact on crop development, plant height, canopy closure, and yield. As planting date was delayed, the time required for crop development decreased from 122 total days for plantings on 25 March to 83 days for plantings on 15 July. For plantings after 2 June, plant height decreased by 1.1 cm per day. Canopy closure decreased by 1.01% per day after 27 May. Soybean yield decreased 26.7 kg/ha per day when soybean was planted after 20 April. This research demonstrates the importance of early planting dates for soybean producers in the southern US to ensure profitability by maximizing yield potential.

**Keywords:** soybean; planting date; crop development

## 1. Introduction

Soybean (*Glycine max* (L.) Merr.), is one of the most important commodities worldwide, and is planted on over 6% of land dedicated to agriculture [1]. Soybean seeds are used in a variety of ways, from protein meal to feeding livestock to vegetable oil. Soybean production increased from approximately 17 million metric tons in 1960 to 230 million metric tons by 2008 [1]. Multiple factors have influenced this increase, including higher yield potential from breeding and major changes in management strategies.

Soybean production has increased in the southern USA, from 4.8 million hectares in 2006 to 5.9 million hectares in 2014 [2]. This change is largely due to commodity prices fluctuating on a

yearly basis. With soybean being planted on 45% of the area dedicated to agronomic crop production, it has become a major commodity in this region [2]. This increase in production has resulted in soybean being planted over a wider period (late March to late June) compared to the early 2000s, when plantings typically occurred from early April to mid–May, but some plantings were made in June for double–cropping situations.

In the early 2000s, the Early Soybean Production System (ESPS) was introduced into the southern region of the USA as a way to increase yield potential. Recommended practices for the ESPS include planting Maturity Group IV or early–maturity Group V soybean cultivars from late March through late April [3]. Prior to these recommendations, growers planted late–maturity Group V and maturity Group VI soybean cultivars during May and later [3]. The ESPS was developed as a method to avoid drought and high temperatures during the reproductive stages, which can stress soybean crops and decrease yield potential [4]. Currently, growers plant the same maturity groups throughout the entire planting window. In 2003, 36% of the soybean crop across the mid–southern states of Arkansas, Louisiana, and Mississippi was planted before 1 May, 28% between 1 May to 1 June, and 36% after 31 May [3]. Not only does the ESPS provide for greater yield potential, but it can also aid in weed control and insect pest management [5,6].

With grain hectares in the region increasing in the past decade, it has become more difficult to plant all the soybean hectares on large farms in an early window. Most growers utilize the same equipment to harvest soybean, corn (*Zea mays* L.), and grain sorghum (*Sorghum bicolor* (L.) Moench). When soybeans remain in the field for extended periods of time after reaching maturity, it is common for pods to shatter, resulting in yield loss [7]. Many growers now utilize planting date and maturity level to manage timely harvest of their crops and to plan for the crops maturing at different intervals to minimize in–field yield loss, and as a plan for harvest equipment efficiency. However, growers are still predominately utilizing indeterminate–maturity Group IV and V soybean cultivars.

Because planting dates for soybean range from late March through mid–June, knowing the yield potential of soybean based on planting date alone, and how planting date affects other agronomic aspects of soybean production, such as days to maturity, plant height, and canopy closure is important. The objective of this study was to determine the impact of planting date on days to multiple growth stages, plant heights, canopy closure, and yield, and to narrow down the planting window within which when soybean growth and development is optimal in order to maximize yield potential.

## 2. Materials and Methods

### 2.1. Location and Site Descriptions

Soybeans were planted on seven dates at the R.R. Foil Experiment Station in Starkville, MS and the Delta Research and Extension Center in Stoneville, MS in 2013 and 2014 to evaluate how planting date affected plant development and yield. The experiment was arranged as a split plot within a randomized complete block design. A split–plot design was used to manage harvest within each planting date. The main–plot factor was planting date. Planting dates ranged from the third week of March to the second week of July, with approximately two to three weeks between each planting (Table 1). The subplot factor was cultivar. At each planting date, two different indeterminate cultivars, a Maturity Group IV cultivar (Asgrow® 4632, Monsanto Company, St. Louis, MO, USA) and a Maturity Group V cultivar (Asgrow® 5332, Monsanto Company, St. Louis, MO, USA) were planted at each location to represent the most common maturity groups planted in Mississippi. Soybean seed at all plantings were treated with the Acceleron® seed treatment package (Monsanto Company, St. Louis, MO, USA), consisting of pyraclostrobin, metalaxyl, and fluxapyroxad fungicides and imidicloprid insecticide.

**Table 1.** List of planting dates (Julian dates) for Maturity Group IV cultivar and V cultivar soybeans in Starkville and Stoneville, Mississippi, during 2013 and 2014.

| | Starkville, MS | | Stoneville, MS | |
|---|---|---|---|---|
| | **2013** | **2014** | **2013** | **2014** |
| **Planting Date** | **Date of Planting (Julian Date)** | **Date of Planting (Julian Date)** | **Date of Planting (Julian Date)** | **Date of Planting (Julian Date)** |
| 1 | 3/28 (87) | 3/21 (80) * | 4/8 (98) * | 3/25 (84) |
| 2 | 4/17 (107) | 4/17 (107) | 4/15 (105) | 4/16 (106) |
| 3 | 5/9 (129) | 5/8 (128) | 5/8 (128) | 5/12 (132) |
| 4 | 5/30 (150) | 6/4 (155) | 5/30 (150) | 6/3 (154) |
| 5 | 6/13 (164) | 6/16 (167) | 6/13 (164) | 6/13 (164) |
| 6 | 7/1 (182) | 6/30 (181) | 7/1 (182) | 6/30 (181) |
| 7 | 7/15 (196) | 7/16 (197) | 7/15 (196) | 7/16 (197) * |

* Soybean plant densities were not sufficient for these planting dates because of adverse environmental conditions during the germination and emergence period.

Subplot were four rows wide by 4.3 m long, with a row width of 0.96 m in Starkville and 1.01 m in Stoneville, and were planted at density of 44,515 seeds per hectare. There were four replications for each planting date and cultivar per location. Both locations used conventional tillage practices. Conventional tillage practices consisted of disking in the fall and bed preparation in the early spring for furrow irrigation. A foliar application of azoxystrobin (Quadris®, Syngenta Crop Protection, Greensboro, NC, USA) at 0.1 kg ai/ha was applied to all experimental units at the R3 growth stage. The soil type at the Starkville location was a Marietta fine sandy loam (Fine–loamy, siliceous, active, thermic Fluvaquentic Eutrudepts) and a Bosket very fine sandy loam (Fine–loamy, mixed, active, thermic Mollic Hapludalfs) at the Stoneville location. Both locations were fertilized based on soil tests conducted the previous fall. No fertilizer was needed in either year in the Starkville location but $P_2O_5$ at 45 kg/ha and $K_2O$ at 67 kg/ha was applied in the Stoneville location both years. Irrigation practices at both locations consisted of furrow irrigation. Each irrigation event lasted until water reached the tail–ditch on all rows, approximately 0.01 hectare meters of water, and were irrigated weekly from the R1 through R6 growth stages. All experimental units were treated for insects based on action threshold recommendations in the Insect Control Guide for Agronomic Crops [8] published by the Mississippi State University Extension Service. The only pests that reached action thresholds were a complex of stink bugs consisting of green stink bug (*Acrosternum hilare*, Say), southern green stink bug (*Nezara viridula*, L.), and brown stink bug (*Eushistus servus*, Say), along with soybean looper (*Chrysodeixis includens*, Walker) and bean leaf beetle (*Cerotoma trifurcate*, Forster).

*2.2. Measurements*

Insect densities were measured weekly by conducting 25 sweeps per experimental unit using a standard 38.1 cm diameter sweep net from the R1 (first flower) through the R7 (first brown pod) growth stages [9]. Growth stages were recorded weekly from the R1 growth stage until the R7 growth stage. Final plant heights were recorded at the R6 (pod containing full size green beans) growth stage [9], using rulers to measure the length between the base of the plant to the highest node on the plant. Canopy closure was also recorded on the same day as plant heights by measuring average centimeters of open canopy between rows at ten locations per experimental unit, and then converting this number to a percentage of closed canopy based on row spacing. Days to emergence were measured based on daily visual observations, starting every day after planting until soybeans reached cotyledon growth stage. Experimental units were harvested independently by planting date and cultivar at the R8 growth stage when moisture content was approximately 13%, using a two row Kincaid (Massey Ferguson) 8XP plot combine, equipped with a 6 foot platform header and data collection system that tested for seed moisture and test weight of each individual experimental unit. To calculate yield, moisture for all experimental units was corrected to 13%.

*2.3. Statistical Analysis*

Data for days to emergence, R1, and R7 were analyzed with analysis of variance (PROC GLIMMIX. Version 9.4, SAS Institute Inc., Cary, NC, USA). Fixed effects were planting date, maturity group, and the interaction between planting date and maturity group. Random effects included site–year (location by year), replication nested within site–year, replication by planting date nested within site–year, and replication by planting date by maturity group nested within site–year. Mean separations were based on Fisher's protected LSD (least significant difference) with an alpha level of 0.05. Data for plant heights, percent canopy closure, and yield were analyzed using piecewise regression analysis [10] in PROC GLIMMIX so that a random statement that included site–year, replication nested within site–year, replication by maturity group nested within site–year, and replication by planting date by maturity group nested within site–year, could be utilized. With piecewise regression, two slopes were produced along with an exact breakpoint of where the slopes changed. Piecewise regression was chosen over other regression models because it was the best fit, although other models were considered in preliminary analysis. For final plant heights, percent canopy closure, and yields, there was no significant effect for cultivar or interaction between planting date and cultivar for total insect pests ($p = 0.35$), plant heights ($p = 0.21$), canopy closure ($p = 0.84$), or yield ($p = 0.86$); therefore, data were pooled across cultivars at each planting date. In both years and at both sites, all dependent variables responded similarly; therefore, data were pooled across years and sites and were considered random (site–year). No effect of cultivar was observed for any year or planting date for any of the measurement taken; therefore, all analysis was pooled across cultivars for each planting date and year. For all data, planting date is represented by the Julian date for the date of planting. Additionally, the correlation between all of the dependent variables measured and soybean yields were determined (PROC CORR, Version 9.4, SAS Institute, Cary, NC, USA).

The first plantings in both 2013 (Stoneville) and 2014 (Starkville) (Julian day 80–98) had less than sufficient plant densities due to cold temperatures and moisture, whereas Maturity Group V plots for the last planting date (Julian day 196–197) did not have sufficient soil moisture following planting, leading to poor emergence and insufficient plant density during 2014 (Stoneville). Observations for these planting dates were excluded from all analyses. Experimental units for all other planting dates had plant densities that ranged from 37,200 (85% of the seeding rate) plants per hectare to 41,800 (95% of the seeding rate) plants per hectare, which were deemed sufficient for the experiments.

## 3. Results and Discussion

*3.1. Planting Date Effects on Insect Pressure*

Differences were observed for total insect pests based on planting date ($p < 0.01$). The early July (Julian day 181) and mid–July (Julian day 196) plantings received significantly higher densities of total insect pests compared to all other plantings (Figure 1). Of the insect pests reaching action thresholds, the stink bug complex received the most threshold applications. Stink bugs reached action thresholds in all plantings except the mid–April (Julian day 105) and mid–May (Julian day 128) plantings. Soybean looper and bean leaf beetle were only at action threshold densities for plantings starting in early June (Julian day 150).

Means with the same letter are not significantly different according to Fisher's protected LSD test ($\alpha = 0.05$).

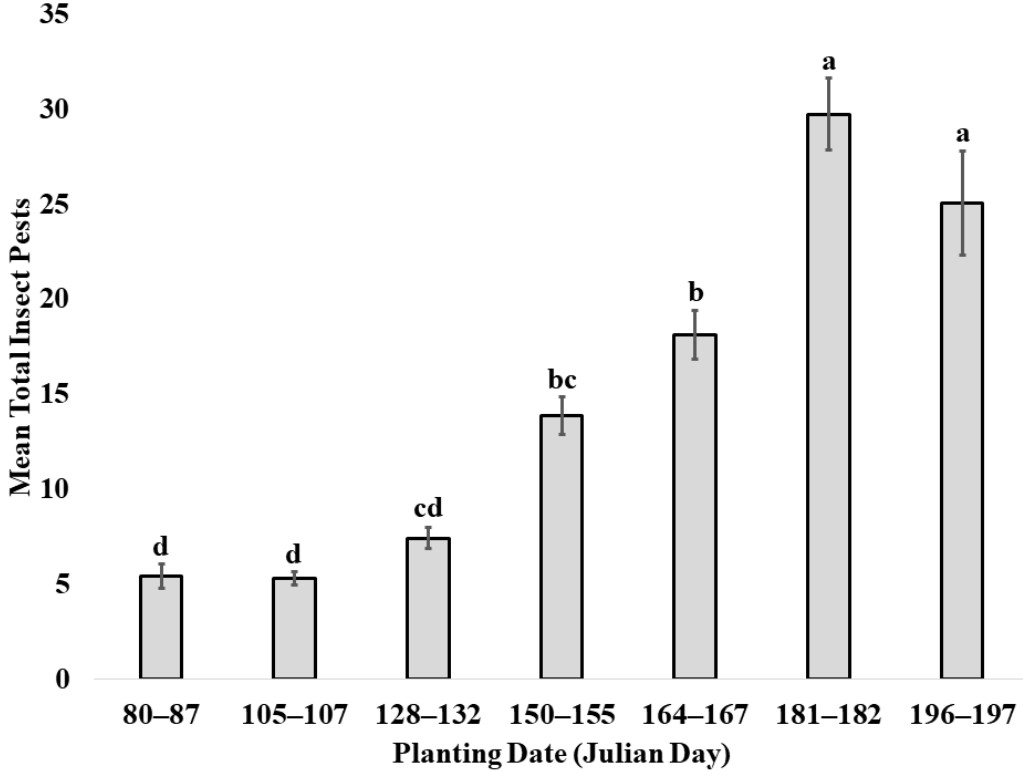

**Figure 1.** Mean total insect pests (±SEM [†]) based on planting date. [†] Standard error of the mean.

### 3.2. Planting Date Effects on Growth Stage

The number of days to selected growth stages varied among the seven planting dates in this study. Differences were observed for emergence ($p < 0.01$), vegetative period ($p = 0.01$), and reproductive period ($p < 0.01$) among planting dates. Soybeans planted before mid–May (Julian Day 132) took longer to emerge than soybeans planted after that time (Table 2). A similar trend was observed for the number of days until the vegetative period, where the significant change in days until the vegetative period did not occur until soybeans were planted in mid–May (Julian Day 132), although the early June (Julian Day 155) planting did not significantly differ from any planting date. Plantings in mid–March (Julian Day 80) took significantly more days to reach the reproductive period compared to all other plantings (Table 3). Significantly fewer days to reach the reproductive period were observed for the early July (Julian Day 181) and mid–July (Julian Day 196) plantings compared to all other plantings (Table 2).

**Table 2.** Total days from planting to reach selected growth stages averaged over cultivars.

| Planting Date (Julian Day) | Average Number of Days to Growth Stage (± SEM) [†] | | |
| --- | --- | --- | --- |
| | Emergence | Vegetative Period | Reproductive Period |
| 1 (80–98) | 10 (1) a | 36 (1) a | 78 (5) a |
| 2 (105–107) | 9 (0) a | 37 (2) a | 66 (4) b |
| 3 (128–132) | 9 (1) a | 30 (2) b | 63 (1) bc |
| 4 (150–155) | 6 (0) b | 33 (2) ab | 58 (4) bc |
| 5 (164–167) | 6 (1) b | 31 (2) b | 56 (1) c |
| 6 (181–182) | 6 (0) b | 32 (1) b | 46 (2) d |
| 7 (196–197) | 5 (0) b | 30 (3) b | 46 (3) d |
| *p*–value | <0.01 | <0.01 | <0.01 |
| LSD | 1.53 | 1.96 | 3.68 |

Means within column followed by the same letter are not significantly different according to Fisher's protected LSD test ($\alpha = 0.05$). [†] Standard error of the mean.

**Table 3.** Linear regression for plant height, canopy closure, and yield for multiple planting dates, and break points (BP) [†] for piecewise regression.

| Data Rated | BP [†] | Intercept (±SEM) | Estimate (±SEM [††]) | t–Value | df | p–Value |
|---|---|---|---|---|---|---|
| | | | **Linear Term** | | | |
| Plant Heights | 153 | | | | | |
| Before BP | | 51.7 (4.2) | 0.3 (0.03) | 13.4 | 369.3 | <0.01 |
| After BP | | 278.4 (6.0) | −1.1 (0.03) | −34.6 | 429.0 | <0.01 |
| Canopy Closure | 147 | | | | | |
| Before BP | | 99.9 (1.9) | −0.1 (0.02) | −4.7 | 334.0 | <0.01 |
| After BP | | 239.5 (4.3) | −1.0 (0.02) | −51.3 | 406.7 | <0.01 |
| Yield | 110 | | | | | |
| Before BP | | 1125.7 (861.0) | 34.3 (6.7) | 5.1 | 5.5 | 0.03 |
| After BP | | 7709.5 (422.1) | −26.6 (1.9) | −13.9 | 123.6 | <0.01 |

[†] Break point: Julian day at which the slope changed. [††] Standard error of the mean.

### 3.3. Planting Date Effects on Plant Height

Final plant heights ranged from 38.1 cm to 142.2 cm with an average height of 83.8 cm. A relationship was observed between planting date and plant height (Table 3). Based on the piecewise regression, a change in slope was observed for the early June plantings (Julian Day 153), with soybeans planted on this date achieving the greatest plant height (Figure 2). For planting dates from 25 March (Julian Day 84) to 2 June (Julian Day 153), plant heights increased by 0.3 (± 0.1) cm per day. For planting dates from 2 June (Julian Day 153) to 16 July (Julian Day 197), plant heights decreased by 2.7 (± 0.2) cm per day.

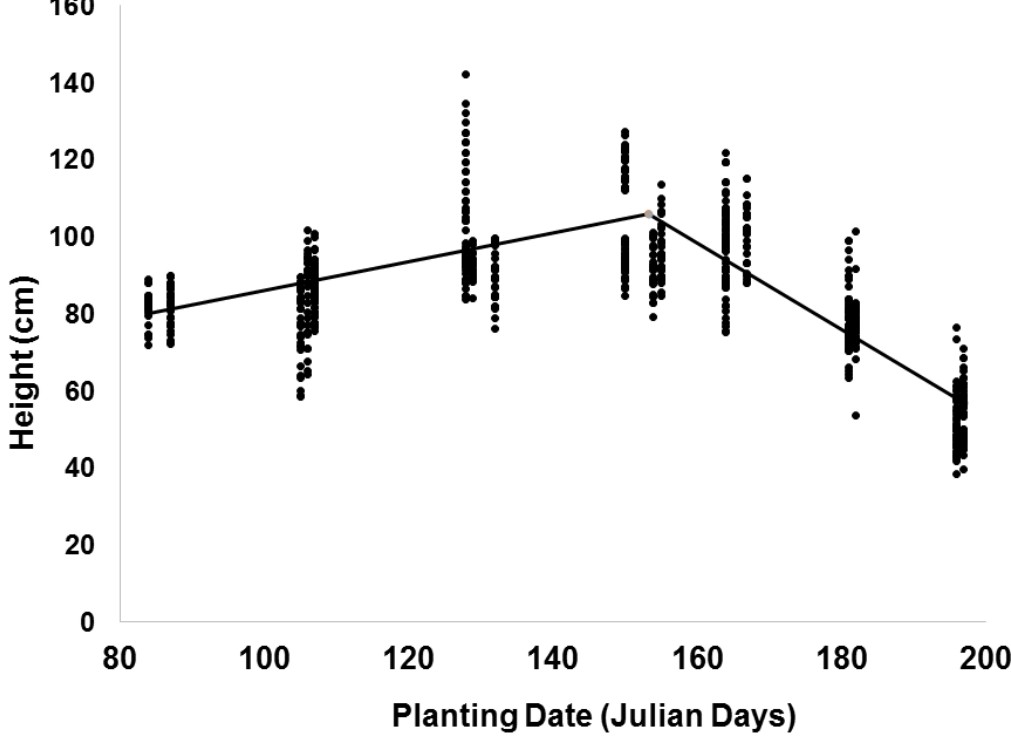

**Figure 2.** Piecewise regression for plant height by Julian days. The equation for the portion before the break point was y = 0.14x (± 0.01) + 20.35 (± 1.62) with a *p*–value of <0.01. The equation for the portion after the break point was y = −0.44x (± 0.01) + 109.59 (± 2.35) with a *p*–value of <0.01. The break point between the two slopes was 153 Julian days.

### 3.4. Planting Date Effects on Canopy Closure

Percent canopy closure ranged from 100% closed canopy to 31.6% closed canopy with an average of 75.2%. A relationship was observed between planting date and percent canopy closure (Table 3). Based on the piecewise regression, a change in the slope occurred at the late May (Julian Day 147) planting date, with percent canopy closure decreasing at later planting dates (Figure 3). Peak percent canopy closure occurred for the 25 March (Julian Day 84) plantings. For planting dates from 25 March (Julian Day 84) to 27 May (Julian Day 147), percent canopy closure decreased by 0.1% (± 0.1%) per day, but percent canopy closure decreased by 1.0% (± 0.1%) per day for planting dates after 27 May (Julian Day 147). Early planting dates resulted in nearly complete canopy closure by R5, while canopy closure for soybean planted after 27 May (Julian Day 147) never exceeded 89% canopy closure.

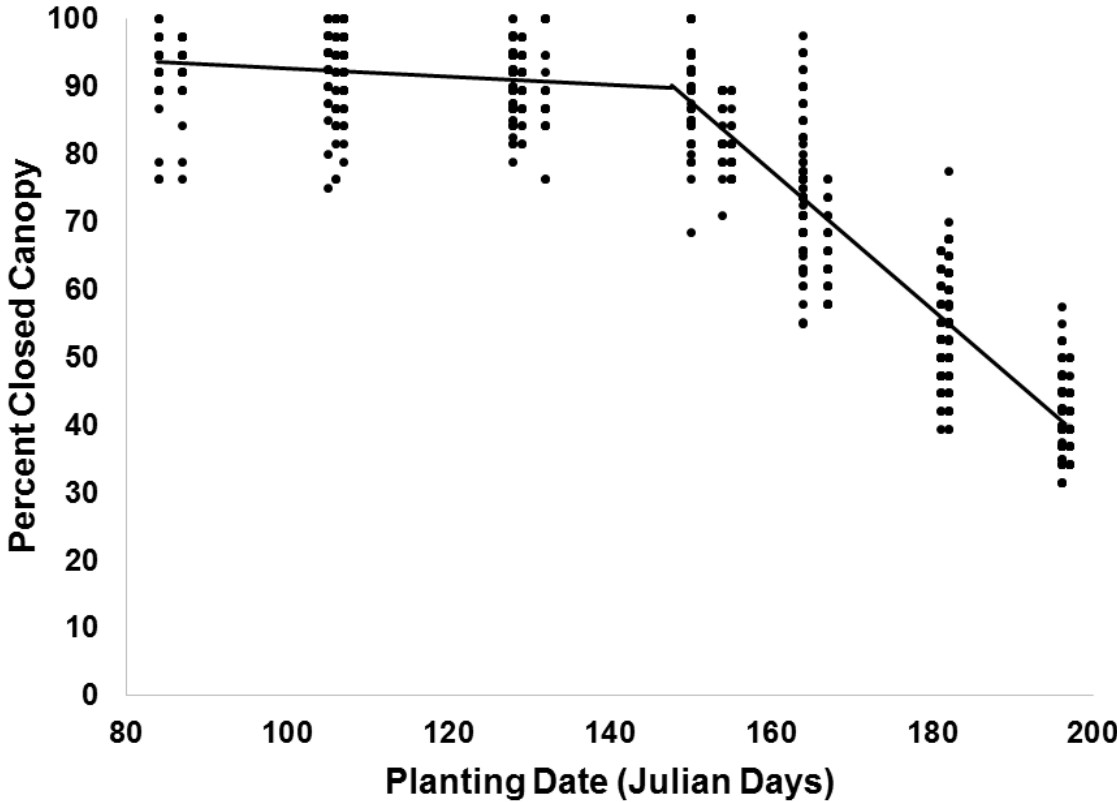

**Figure 3.** Piecewise regression for percent canopy closure. The equation for the portion before the break point was y = −0.07x (± 0.02) + 99.89 (± 1.85) with a *p*–value of <0.01. The equation for the portion after the break point was y = −1.01x (± 0.02) + 239.54 (± 5.97) with a *p*–value of <0.01. The break point was 147 Julian days.

### 3.5. Planting Date Effects on Yield

Yield ranged from 6321 kg/ha to 1372 kg/ha with an average yield of 3564 kg/ha. A significant relationship was observed between planting date and yield (Table 3). Based on the piecewise regression, a change in slope was observed after mid–April plantings (Julian Day 110), the planting date where yield was maximized (Figure 4). For planting dates from 25 March (Julian Day 84) to 20 April (Julian Day 110), yield increased by 34.3 (± 6.7) kg/ha per day, while yield decreased by 26.6 (± 1.9) kg/ha per day for planting dates from 20 April (Julian Day 110) to 16 July (Julian Day 197).

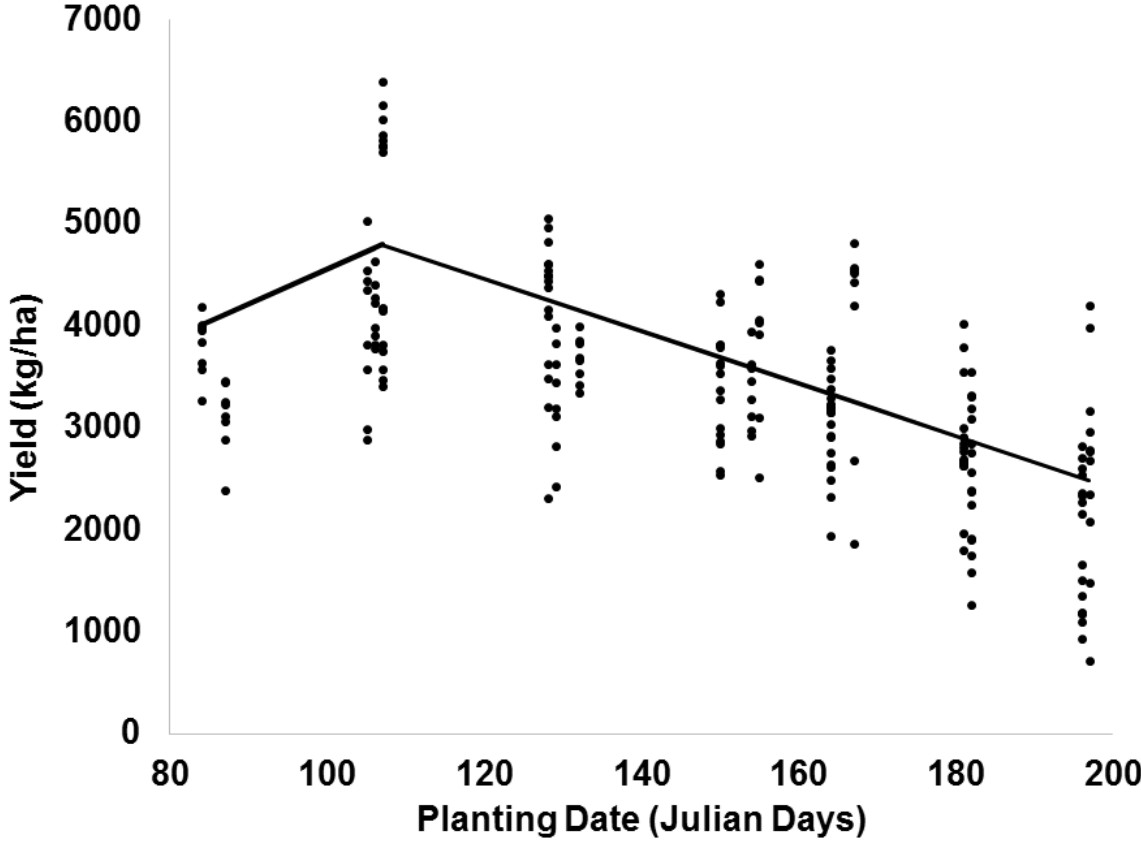

**Figure 4.** Piecewise regression for yield. The equation for the portion before the break point was $y = 0.51x (\pm 0.09) + 16.74 (\pm 12.81)$ with a *p*–value of <0.01. The equation for the portion after the break point was $y = -0.39 (\pm 0.03) + 114.67 (\pm 6.28)$ with a *p*–value of 0.03. The break point was 110 Julian days.

*3.6. Plant Height, Canopy Closure, Days to Growth Stages, and Insect Pressure Impact on Yield*

Plant heights, canopy closure, days to growth stages, and insect pressure all had a significant ($p < 0.01$) positive correlation with soybean yield except total insects, which had a significant ($p < 0.01$) negative correlation with soybean yield. In general, yields were greater with greater days to plant emergence ($r = 0.29$), greater days to R1 ($r = 0.49$), greater days to R7 ($r = 0.57$), greater percentage of canopy closure ($r = 0.57$), and greater plant heights ($r = 0.40$). In terms of plant height, lodging occurred with the tallest plants in some situations and yields could have been negatively impacted when that occurred. Yields were negatively impacted with greater insect pressure ($r = -0.41$). These results suggest that all of these factors are important for maximizing soybean yields, but no single factor was strongly correlated with yield based on the correlation coefficients. This is most likely because relationships between all factors and yield were not linear.

## 4. Recommendations for Soybean Production Based on Planting Date

The data from the studies conducted in 2013 and 2014 are consistent with multiple studies indicating that as planting is delayed, the developmental time of soybean crops is more rapid, potentially impacting plant height, canopy closure, and yield [11–14]. With increased developmental time, plant heights can increase leading to a greater chance of lodging [15]. Canopy closure is also highly dependent on developmental time; without the proper amount of time, the canopy will not fully close, which can hinder photosynthesis and in turn decrease yield. Yield is directly impacted by both canopy closure and plant height, and without adequate developmental time, yield potential will decrease in soybean crops [11,14,15].

Plant height increases with planting date when soybeans are planted early, but starts to decrease with plantings after early June. Previous research indicated a similar response in that soybean plants planted in June reached a greater plant height when compared to those planted before June, resulting in greater lodging [16,17]. The day length and temperature for early plantings and late plantings differ greatly and final plant heights may be affected by these factors [12].

Lodging was only observed in plantings between mid–May and mid–June, which could potentially cause direct yield losses. Lodging was due in large part to the height the plants reached, which amplified the effect that wind had on the plants. After the experiment was conducted, it was discovered that the Asgrow® 5332 cultivar is prone to lodging, but the same amount of lodging was observed in the Asgrow® 4632 cultivar. Growers should try to plant early in the planting window to avoid these losses, or choose cultivars with less vegetative growth potential when planting in this window.

Canopy closure was not affected by planting date until soybeans were planted after late May. Taylor [18] observed that a fully closed canopy can help maximize photosynthesis. Furthermore, a fully closed canopy can aid in weed control due to reduced sunlight reaching the soil surface [6]. This may assist with the management of herbicide–resistant weeds that are common in the mid–South. Nordby [19] reported lower Amaranthus palmeri L. densities in soybean with closed canopies compared to those with more open canopies. Knowing this, and the current state of resistant weed control issues throughout Mississippi, early plantings of soybean where the canopy reaches full closure will aid in controlling resistant weed populations. As the canopy closes, it blocks sunlight from reaching the soil surface, which in turn can prevent or inhibit the emergence and growth of these problematic weeds [6].

The lack of complete canopy closure in late plantings could be attributed to less time being available for vegetative growth. Similar results have been observed where later planted soybean required fewer total days to reach maturity [13]. The fewer days for soybean plants planted later to reach maturity can be attributed to the shortening of day length throughout the growing season. In a similar study, Egli and Bruening [20] observed that May plantings of soybean resulted in more biomass than those that were planted in June. Yield is the most important factor to a soybean farmer and was also highly affected by planting date. After 20 April, growers start to lose yield potential, which directly affects their potential profit for the growing season. This becomes an even greater factor in times where commodity prices are in decline and the cost of growing soybean is increasing. In this two–year study, yield was maximized on 20 April. These experiments were conducted in intensively managed, irrigated growing environments, and results may not be representative for other environments. In regions without irrigation, or with limited rainfall, planting date could have an even larger impact on yield potential of soybean [21]. The increase in yield for the earlier plantings was most likely due to optimal growing conditions with cooler temperatures, compared to later plantings with high temperatures and little rainfall. With that being said, results from the 2013 and 2014 experiments were similar to those from studies conducted in Oklahoma during 2009 and 2010, where plantings before mid–May resulted in greater yields than later plantings (after mid–May) of soybean [22]. In regional analyses across the mid–southern portion of the United States, soybeans planted after early–May had significantly lower yields than soybeans planted before early May [20,23]. Early plantings can result in weed suppression and can decrease the likelihood of large insect infestations. Similarly to our results, observations have been made in the mid–South that late plantings of soybean received large densities of multiple insect pests [24,25]. Studies have also shown early plantings of soybean acting as harbors for stink bug [26,27], similar to results observed in the current study.

Seed quality was not measured in this study, but it is common to see a decrease in quality as soybeans are planted later due to adverse rainfall in the late reproductive stages, and potentially from late–season stink bug feeding. Soybeans planted too early or too late may be negatively impacted by environmental conditions that are not ideal for quality stand emergence. This was observed in the current study where poor seedling emergence was observed for the first planting date at two site–years and for the last planting date at one site–year. These data suggest that the optimum planting dates for soybean in Mississippi are between 10 April and 1 May, and that this planting window will help

with all aspects of soybean agronomics, including stand quality, plant heights, canopy closure, insect pressure, and maximizing yield.

**Author Contributions:** A.L.C. and J.G. were the co-advisors for this project and oversaw the experiment, as well as, provided input on experimental design and secured funding for this project. D.R.C., F.R.M., and J.T.I. were critical in data analysis and provided additional input on crop management throughout the growing season. N.R.B. conducted this experiment as part of his dissertation work. All authors have read and agreed to the published version of the manuscript.

**Funding:** This research was funded by the Mississippi Soybean Checkoff administered through the Mississippi Soybean Promotion Board, grant number 58–2016.

**Conflicts of Interest:** The authors declare no conflict of interest.

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
