# Peer review of "Effects of Planting Date for Soybean Growth, Development, and Yield in the Southern USA"

_agronomy, doi:10.3390/agronomy10040596_

Round 1

Reviewer 1 Report

This manuscript investigated the effects of planting date on soybean agronomic traits in Mississippi state, US and recommended that the current optimum date is between 10 April to 1 May for growers in this region. I enjoyed reading this paper and finding idea interesting. This is a nice paper.

L37: It had better to use SI units instead of acres for readers other than the US and UK.

L88: It had better to use SI units instead of feet and inches.

Author Response

Response to Reviewer 1 Comments

Comment 1: L37-It had better to use SI units instead of acres for readers other than the US and UK.

Response 1: All units have been changed to metric.

Comment 2: L88-It had better to use SI units instead of feet and inches.

Response 2: All units have been changed to metric.

Reviewer 2 Report

Line 36-63
This section is very focused on the situation in Mississippi & the USA.
Would it be better with more international focus?

Line 67-68
The objective sentence is too brief.
It does not match the subsequent results?
Also the word ‘impact’ is vague. What do you mean??
How about writing a hypothesis or two?
Show us what your thinking is and what you are trying to test in a clear way.

Line 70-106
This section is written in an awkward and poorly ordered manner.
The sentences from line 85 to 87 should go earlier.
It is confusing to read.

Line 87-89 50 ft, 38 inches, acres??
An international scientific paper should use standard metric units.
Indeed later in the paper
Line 197 you use cm !!
This is inconsistent and not helpful.

Line 114
How is this canopy measurement done? With a ruler?
Not sure of how accurate it would be !

Line 144 – what is sufficient?

Line 150 – table 2
This table is too large.
What is the point of this table? What effects are you showing?

Line 154
Insect pressure.
Is this an impact?
Not a clear stated objective?

Line 166 – fig. 1 was there a cultivar effect?

Line 170 – table 3.
This table is too large, is all the detail needed?
What is the point ? what are the effects?

Line 186 – lodging.
Was any observed in the study?

Line 192 – table 4
What was there a cultivar effect?

Line 206 – ‘vary’…..
OK, but was there a significant difference?

Line 208 – table 5
Why do we need this stat. output table?
How valid is the piecewise regression?
What is the justification for this model?
I am not sure how valid it is.
How meaningful or valid are the BP days??

Fig. 2 and 3 – how valid/ sensible is the piecewise regression?
I am not convinced by it.
Is there a biological justification for it?

Line 245 – mixing units
Kg/ ha and bu/ acre??
This is confusing and not helpful at all !

Line 257 which variables?

Line 261 – lodging – please explain what the effect of lodging was on yield.
This is not clear enough.

Line 270 – what about soil temperature?

Line 279 – lodging took place………
OK, but what was the size of the effect?

Line 282-3 – this is not a very helpful sentence.
So there is no difference between the cultivars………
But you think there should be???

Line 291 – yield…………
What about quality??

Line 294 – price to grow soybean………….
There were no crop economics presented in this study……

Line 295 – optimize?
How? what sort of optimization?

Line 301-2
How does that link to any of the objectives??

Author Response

Response to Reviewer 2 Comments

Comment 1: Line 36-63

This section is very focused on the situation in Mississippi & the USA.

Would it be better with more international focus?

Response 1: The introduction was reworded to show the importance of soybean production on an international scale instead of being as focused on one state in the USA. The abstract was slightly altered as well to reflect some of these changes.

Comment 2: Line 67-68

The objective sentence is too brief.

It does not match the subsequent results?

Also the word ‘impact’ is vague. What do you mean??

How about writing a hypothesis or two?

Show us what your thinking is and what you are trying to test in a clear way.

Response 2: More wording was added to the objective statement to better define what the authors were attempting with these studies.

Comment 3: Line 70-106

This section is written in an awkward and poorly ordered manner.

The sentences from line 85 to 87 should go earlier.

It is confusing to read.

Response 3: This section was reworded and rearranged slightly to aid in readability.

Comment 4: Line 87-89 50 ft, 38 inches, acres??

An international scientific paper should use standard metric units.

Indeed later in the paper

Response 4: All units have been converted to metric units throughout the paper

Comment 5: Line 197 you use cm !!

This is inconsistent and not helpful.

Response 5: All units have been converted to metric units throughout the paper

Comment 6: Line 114

How is this canopy measurement done? With a ruler?

Not sure of how accurate it would be !

Response 6: This was conducted with a ruler. The authors agree that it may not be the most accurate measurement, but it was done consistently from plot to plot and the differences should be relative among planting dates.

Comment 7: Line 144 – what is sufficient?

Response 7: 85% or greater of the planted seed was considered sufficient. This was addressed in the next sentence.

Comment 8: Line 150 – table 2

This table is too large.

What is the point of this table? What effects are you showing?

Response 8: The original reasoning for having this table is to show plant stands for each planting date and year, but the authors agree, this table did not strengthen the paper and has been removed.

Comment 9: Line 154

Insect pressure.

Is this an impact?

Not a clear stated objective?

Response 9: This was not an objective of the paper, but insect pressure was monitored weekly, and there was a large difference between planting dates with respect to insect pest densities, so it was included in the paper.

Comment 10: Line 166 – fig. 1 was there a cultivar effect?

Response 10: There was no cultivar effect for any of the measurements taken, and all analysis was pooled across cultivars. This is now addressed in the materials and methods section.

Comment 11: Line 170 – table 3.

This table is too large, is all the detail needed?

What is the point ? what are the effects?

Response 11: This table is not need and has been deleted. The overall story of the paper is the impact planting date has on agronomic factors of soybean and not insects. The figure should suffice, and the insect pests that reached threshold are addressed in the materials and methods section.

Comment 12: Line 186 – lodging.

Was any observed in the study?

Response 12: Minimal lodging was observed in this study, less than 10%, and was sporadic and not even among a planting date or cultivar. We do elude to the fact that this could have impacted yield, but no formal measurements of lodging were recorded. If the reviewer wishes, we can remove our statements about lodging impacting yield.

Comment 13: Line 192 – table 4

What was there a cultivar effect?

Response 13: There were no cultivar effects, and this is no addressed in the materials and methods section.

Comment 14: Line 206 – ‘vary’…..

OK, but was there a significant difference?

Response 14: Vary was changed to differ to reflect that day length and temperature were different among the early and late plantings.

Comment 15: Line 208 – table 5

Why do we need this stat. output table?

How valid is the piecewise regression?

What is the justification for this model?

I am not sure how valid it is.

How meaningful or valid are the BP days??

Response 15:

This table was used to simplify the regression equations and the statistics associated with the equations. We felt this table would be easier to interpret than having the equations and statistics in the text.

Piecewise regression is a valid analysis for this data set. It has been used on numerous other data sets similar to ours (citations below). The reason we used this analysis is because it was a better fit than a linear or quadratic model. The linear model did not show the yield loss being as great as was observed for plantings after 1 May. With a quadratic model, the peak of the curve is pushed past the point at which yield was actually maximized, showing the peak between early and mid-May. We know from experience this isn’t the case for soybean in Mississippi.

The second purpose of using the piecewise, was to determine the exact day at which yield (and all other agronomic factors) is maximized. This analysis is designed to determine the break point (BP) at which a slope changes. The use of this analysis helped us determine the day at which all of the factors tested were maximized.

Choudhury, Askar H., James R. Jones, and Aslihan D. Spaulding. "Association of rainfall and detrended crop yield based on piecewise regression for agricultural insurance." Journal of Economics and Economic Education Research 16.2 (2015): 31.

Singh, Ramesh P., et al. "Crop yield prediction using piecewise linear regression with a break point and weather and agricultural parameters." U.S. Patent No. 7,702,597. 20 Apr. 2010.

Blanco, Flávio Favaro, et al. "Growth and yield of corn irrigated with saline water." Scientia Agrícola 65.6 (2008): 574-580.

Schmitz, Andrew, and Fangyi Zhang. "The Dynamics of Sugarcane and Sugar Yields in Florida: 1950–2018." Crop Science 59.5 (2019): 1880-1886.

Comment 16: Fig. 2 and 3 – how valid/ sensible is the piecewise regression?

I am not convinced by it.

Is there a biological justification for it?

Response 16: Please refer to response 15.

Comment 17: Line 245 – mixing units

Kg/ ha and bu/ acre??

This is confusing and not helpful at all !

Response 17: All units have been converted to metric units.

Comment 18: Line 257 which variables?

Response 18: All variables are now listed instead of just stating all variables

Comment 19: Line 261 – lodging – please explain what the effect of lodging was on yield.

This is not clear enough.

Response 19: Wording was altered slightly to suggest that lodging could have impacted yield. No formal measurement of lodging was recorded, so no analysis can be conducted on lodging effects.

Comment 20: Line 270 – what about soil temperature?

Response 20: Soil temperature was different but this was not recorded for every planting. It was recorded for the initial planting, to ensure the temperature was warm enough for soybean to germinate.

Comment 21: Line 279 – lodging took place………

OK, but what was the size of the effect?

Response 21: No formal lodging ratings were recorded so no analysis on lodging was conducted. Wording was altered slightly to suggest lodging could impact yield.

Comment 22: Line 282-3 – this is not a very helpful sentence.

So there is no difference between the cultivars………

But you think there should be???

Response 22: At the time this experiment started the lodging differences between the two cultivars was not known. We did not expect to see a difference among the cultivars with respect to lodging, but were trying to determine if planting date effected yield and growth a development differently among the cultivars.

Comment 23: Line 291 – yield…………

What about quality??

Response 23: Quality was not measured in this study. A sentence was added that quality could be impacted from both weather and insect pressure in late planted soybean. We typically do not see quality differences for early planted soybean in our region.

Comment 24: Line 294 – price to grow soybean………….

There were no crop economics presented in this study……

Response 24: We were simply referring to the increase in fuel, chemical, and seed cost we have observed over the past 5-6 years. Some of this has been influenced from weed and insect resistance, which has led to changing to chemical classes that are more expensive. The authors will add wording to reflect this is the reviewer would like.

Comment 25: Line 295 – optimize?

How? what sort of optimization?

Response 25: Optimize has been changed to maximize.

Comment 26: Line 301-2

How does that link to any of the objectives??

Response 26: This sentence was reworded to link back to the objectives.

Reviewer 3 Report

I have attached a short review of specific items.

Go through methods and make sure values are in metric units. Chemical formulas should have appropriate subscript or superscript.

Use experimental units for areas of measurement, not the term plots.

Line 79- change to metalaxyl

Lines 85 to 109. Several entries in English units. I don’t think this is allowed. I think that English units can be added in parentheses following the metric units.

Line 97- subscript in chem formulas needed.

Line 147- Use ‘experimental units’, not ‘plots’- can be confused with locations.

Line 245- one whole number should be sufficient significant figures. Decimal value superfluous.

Line 250- 1.9 is sufficient, the hundredths is not necessary

The work should be valuable for south US soybean farmers.

Author Response

Response to Reviewer 3 Comments

Comment 1: Go through methods and make sure values are in metric units. Chemical formulas should have appropriate subscript or superscript.

Response 1: All units have been converted to metric.

Comment 2: Use experimental units for areas of measurement, not the term plots.

Response 2: All instances where plot was used have been changed to experimental unit

Comment 3: Line 79- change to metalaxyl

Response 3: Changed to metalaxyl

Comment 4: Lines 85 to 109. Several entries in English units. I don’t think this is allowed. I think that English units can be added in parentheses following the metric units.

Response 4: All units converted to metric

Comment 5: Line 97- subscript in chem formulas needed.

Response 5: All ® marks have been superscripted.

Is this what the reviewer was referring to?

Comment 6: Line 147- Use ‘experimental units’, not ‘plots’- can be confused with locations.

Comment 6: Plots were changed to experimental units

Comment 7: Line 245- one whole number should be sufficient significant figures. Decimal value superfluous.

Response 7: Decimal values removed, and only whole number left

Comment 8: Line 250- 1.9 is sufficient, the hundredths is not necessary

Response 8: Hundredths place was removed

Comment 9: The work should be valuable for south US soybean farmers.

Response 9: Thank you, our objective when we started the project was to help growers maximize soybean yields.

Reviewer 4 Report

I have attached a manuscript with a number of comments and suggestions.  My main concern is the way development is dealt with.  The current presentation confounds past stages with current states.  Instead of planting to emergence, planting to R1, and planting to R7, I would like to see planting to emergence, vegetative period (R1 - emergence), and reproductive period (R7 - R1).

Author Response

Response to Reviewer 4

Comment 1: I have attached a manuscript with a number of comments and suggestions.  My main concern is the way development is dealt with.  The current presentation confounds past stages with current states.  Instead of planting to emergence, planting to R1, and planting to R7, I would like to see planting to emergence, vegetative period (R1 - emergence), and reproductive period (R7 - R1).

Response 1: The authors were not able to see the attached manuscript with the comments and suggestions. We would be glad make any additional changes that are not listed above.

Developmental concerns: The authors have changed the headings to reflect the reviewer’s suggestion. The authors agree that this way is less confusing and more clearly defines what the authors were measuring.